# iERM: An Interpretable Deep Learning System to Classify Epiretinal Membrane for Different Optical Coherence Tomography Devices: A Multi-Center Analysis

**DOI:** 10.3390/jcm12020400

**Published:** 2023-01-04

**Authors:** Kai Jin, Yan Yan, Shuai Wang, Ce Yang, Menglu Chen, Xindi Liu, Hiroto Terasaki, Tun-Hang Yeo, Neha Gulab Singh, Yao Wang, Juan Ye

**Affiliations:** 1Department of Ophthalmology, The Second Affiliated Hospital of Zhejiang University, College of Medicine, Hangzhou 310009, China; 2School of Mechanical, Electrical and Information Engineering, Shandong University, Weihai 264209, China; 3Department of Ophthalmology, Kagoshima University Graduate School of Medical and Dental Sciences, Kagoshima 890-8520, Japan; 4Ophthalmology and Visual Sciences, Khoo Teck Puat Hospital, National Healthcare Group, Singapore 768828, Singapore

**Keywords:** deep learning, epiretinal membrane, optical coherence tomography, multi-center

## Abstract

Background: Epiretinal membranes (ERM) have been found to be common among individuals >50 years old. However, the severity grading assessment for ERM based on optical coherence tomography (OCT) images has remained a challenge due to lacking reliable and interpretable analysis methods. Thus, this study aimed to develop a two-stage deep learning (DL) system named iERM to provide accurate automatic grading of ERM for clinical practice. Methods: The iERM was trained based on human segmentation of key features to improve classification performance and simultaneously provide interpretability to the classification results. We developed and tested iERM using a total of 4547 OCT B-Scans of four different commercial OCT devices that were collected from nine international medical centers. Results: As per the results, the integrated network effectively improved the grading performance by 1–5.9% compared with the traditional classification DL model and achieved high accuracy scores of 82.9%, 87.0%, and 79.4% in the internal test dataset and two external test datasets, respectively. This is comparable to retinal specialists whose average accuracy scores are 87.8% and 79.4% in two external test datasets. Conclusion: This study proved to be a benchmark method to improve the performance and enhance the interpretability of the traditional DL model with the implementation of segmentation based on prior human knowledge. It may have the potential to provide precise guidance for ERM diagnosis and treatment.

## 1. Introduction

The epiretinal membrane (ERM) is a pre-retinal proliferative disease, which can result in blurred vision and central vision loss [1,2]. In the 20-year follow-up Beaver Dam Eye Study, the prevalence of ERM was 34.1% [3]. Due to older age being the most common risk factor for ERM, it is likely that the prevalence of ERM will further increase with an aging population. This brings consequential social and economic burdens [4,5].

Optical coherence tomography (OCT), particularly spectral-domain OCT, has been the gold standard for diagnosing ERM in clinical practice due to its higher sensitivity in evaluating ERM compared to other retinal examinations [6,7,8]. OCT can reveal not only the presence of ERM but also the specific severity stage that can assist in guiding disease management [6,9,10]. However, the accurate fine-grading assessment of ERM based on OCT images requires reliable and interpretable analysis methods [11]. The large-scale interpretation of OCT images is deemed to be labor-intensive and time-consuming.

Over the last few years, several studies have shown remarkable results by developing deep learning (DL) systems for automatic retinal disease diagnosis based on OCT images [12,13,14,15]. In the field of ERM diagnosis, those studies prove the ability of DL algorithms to accurately detect ERM from OCT images [16,17,18]. They have the potential to simplify or accelerate OCT image interpretation in clinical practice.

However, several crucial challenges remain. First, no studies to date have tested these algorithms on their classification of ERM into precise severity grading, which could be a guide for referral and treatment. Second, the main drawback of the DL classification system is that it might be difficult for clinicians to trust, thus hindering its practical application. Further effort is required to enhance its interpretability. Third, the OCT data from most research are only obtained from a single commercial device in a single medical center without external datasets to test the generalizability of DL algorithms in a real-world setting.

Thus, to address these gaps, this study developed an interpretable DL system based on multi-center OCT data for ERM severity grading; this was named iERM. It was tested in two independent external datasets. It may have the potential to provide precise guidance in ERM diagnosis and treatment.

## 2. Materials and Methods

The workflow of the iERM system is demonstrated in Figure 1A, which included data collection, image labeling, development of the segmentation model, and development of the classification model.

### 2.1. Data Collection

This was a multi-center, retrospective, cross-sectional study. In total, 4547 OCT B-scans (2606 with ERM and 1968 normal) from four different commercial OCT devices from nine international medical centers were collected for this study.

The primary dataset was collected from 1593 eyes of 1046 ERM patients, who underwent a spectral-domain OCT (Heidelberg, Germany) at the Eye Center at the Second Affiliated Hospital of Zhejiang University, from April 2017 to August 2020 (ZJU dataset). All OCT B-scans were central fovea cross sections. Subjects who had other retinopathies other than ERM that can affect the macula or had overt media opacity were excluded. This study was approved by the Medical Ethics Committee of the Second Affiliated Hospital, Zhejiang University, and complied with the Declaration of Helsinki. The demographic and statistical information of all subjects in the ZJU dataset is shown in Table 1. The other ten datasets followed the same data collection as the ZJU dataset, and each OCT image was collected from different subjects. The examples of OCT data collected from different commercial devices are illustrated in Figure 1B.

This retrospective study is a sub-analysis of data from a clinical study (A New Technique for Retinal Disease Treatment, ClinicalTrials.gov identifier: NCT04718532). Ethical approval for this study was obtained from the Ethics Committee for ZJU-2 (No Y2020–1027). The research adhered to the tenets of the Declaration of Helsinki and the Health Insurance Portability and Accessibility Act.

### 2.2. Image Labeling

In this study, an OCT-based six-stage ERM classification standard was utilized, and the typical case of each stage is shown in Figure 2A [19]. The normal retina was classified in a normal category. In stage 1 of the epiretinal membrane, the mild ERM brought negligible anatomic disruption, while the central fovea had a normal morphology. The outer nuclear layer thickness in the central fovea was markedly thickened, and the central fovea lost its normal morphology in stage 2 of the epiretinal membrane. In stage 3 of the epiretinal membrane, the foveal depression was absent with a continuous ectopic inner foveal layer crossing through the foveal area anomalously. All retinal layers were still identifiable. In stage 4 of the epiretinal membrane, severe retinal thickening and extremely distorted retinal layers were observed. A continuous ectopic inner foveal layer crossing through the foveal area was also identified. Stage 5 of the epiretinal membrane was defined as epiretinal membrane-induced tractional lamellar macular holes or macular pseudo-holes. The label and distribution of all the datasets are shown in Table 1.

For classification, four experienced retinal specialists were recruited to identify the ERM stages of the OCT images independently without communication. The images for the three or four specialists had the same label. The labels of questionable images were confirmed again until full consensus was reached. The label and distribution of all the datasets are shown in Table 2.

For the segmentation work, a five-level stratification of specific layers, namely Layers 1 to 5 was designed according to the key feature of disease progression. The first layer is defined as the morphological curve of the upper boundary of the retina, which is the most distinguishing feature. The second layer was the nerve fiber layer, the ganglion cell layer, and the inner plexiform layer. The inner nuclear layer is the third layer, whose shape is crucial for distinguishing stages 2–4. The fourth layer was the outer plexiform layer and outer nuclear layer, while the last layer was from the external limiting membrane to the pigment epithelium layer. Two experienced retinal specialists were recruited to label the boundaries of these layers. In total, 300 OCT images (50 images for each stage) of the ZJU dataset were labeled to develop the segmentation model.

### 2.3. Data Processing

There were 5 different image specifications of images generated by different OCT devices, namely Heidelberg 1, Heidelberg 2, Crrius, Zeiss, and Nidek. The specifications of images generated by each OCT device are shown in Appendix A. In terms of image mode, Zeiss was a Pseudo-color image, and other images were grayscale images. In the grayscale images, Heidelberg1, Crrius, and Nidek had low grayscale values in the background region and high grayscale values in the retinal region, while Heidelberg 2 is just the opposite. The first step of data processing is to uniformly convert all image modes into grayscale images with low grayscale values in the background region and high grayscale values in the retinal region. The image mode conversion of the Zeiss image and Heidelberg 2 image is shown in Appendix A.

After unifying the image mode, the images of these different devices retained the problems of inconsistent gray value distribution intervals and inconsistent sizes. The gray value distribution histograms are shown in Appendix A. Aiming at the inconsistency of the gray value distribution interval, random changes in brightness, saturation, and contrast were used to enrich the gray value distribution so that the network could be used on images with different gray value distributions. For the problem of inconsistent size, the linear interpolation of the image was unified to the size of 224 × 224, which not only unified the size of the multi-machine images but also conformed to the input size of most classification networks and reduced the computational consumption.

The background area in OCT images, including part of the vitreous chamber and the region below the pigment epithelium layer, was removed to reduce noise. First, a U-Net network was established to distinguish the retina from the background area; then, the maximum connected domain was used to remove the wrong segmentation fragments, and finally, the maximum peripheral rectangle of the retina region was identified [20]. The data augmentation was performed by randomly adjusting the brightness, contrast, and saturation of OCT images and randomly flipping the images in the horizontal direction. Flipping in the vertical direction and rotation were abnegated since they did not follow the natural retinal structure.

### 2.4. Development of the DL System

(1)Segmentation model

A U-Net network was selected as the segmentation model for retinal layer segmentation, which is the most classic and extensively used model in medical image segmentation [20]. U-Net was cascaded by a down-sampling pathway and an up-sampling pathway. The encoder in the down-sampling pathway extracted features from the input image and extracted high-dimensional features from low-dimensional features, while the decoder in the up-sampling pathway restored the high-dimensional feature to the low-dimensional feature, and finally generated the segmentation result consistent with the size of the input image. To pursue more accurate segmentation, skipping the connection into the U-Net to fuse low-dimensional detailed information with high-dimensional abstract information was utilized.

The proposed first layer was to detect the upper boundary of the retina within a width of three pixels to enhance the morphological information. The remaining four layers and background were segmented via the U-Net network. The dataset contained 300 images with the annotations described above and were divided for training, verification, and testing according to the ratio of 7:1:2. The RMSProp optimization algorithm was used with a weight decay of 10^−8^ and a momentum of 0.9 to update the network weights, and a learning rate scheduler with decay of 0.5 per 10 steps was used. The batch size was set to 8, and the maximum epoch was 50. At the end of each epoch, the current network weight was verified, and the weights of the best results were saved.

(2)Classification Model

Three classification models were compared in this study, and the model with the best results was selected as the backbone network of the proposed two-stage system [21,22,23]. The Xian 1 dataset, the Japanese dataset, and part of the ZJU dataset were divided into two independent external test sets and the internal test set without training. The remaining datasets were combined like the internal dataset to develop the DL model. The ratio for training and verification was 7:1. The specific details in terms of the usage of image data for the classification task is shown in Appendix A. The SGD optimization algorithm was used with a momentum of 0.9. The initial learning rate was set to 0.001, and a learning rate scheduler with a decay of 0.5 every 30 steps was used. The batch size was set to 32, and the maximum epoch was 80. At the end of each epoch, the current network weight was verified, and the weights of the best results were saved.

(3)Two-stage System

In the first step of this system, the OCT image was input into the segmentation model and output as a segmentation map. In the second step, the original OCT image and corresponding segmentation map were jointly input into the classification model. The channel concatenation method for merging was utilized. First, the three channels of the original OCT image were standardized, and the values of five channels of the corresponding segmentation map were mapped 0–1 using softmax. The value range of the original image and the segmentation results were consistent after that, which avoided bias with the classification network. Then, the input channel was modified for the first convolutional layer of the classification network to eight to receive three channels simultaneously for the original OCT image and five channels of the corresponding segmentation map as input.

### 2.5. Statistical Analysis

To achieve a comprehensive and objective assessment of our segmentation network, the following metrics are calculated compared with the ground truth: Dice coefficient (Dice), precision, recall, intersection over union (IoU), and average surface distance (ASD). To evaluate the performance of our classification systems, we generated the receiver operating characteristic (ROC) curve by plotting the true positive rate versus the false positive rate. Compared to the ground truth label generated by clinical ophthalmologists, the precision, recall, F1-score, the area under the curve (AUC) of the ROC curve, and total accuracy were calculated.

### 2.6. Model Visualization

To further investigate the classification model, a class activation map for the last convolutional layer of the classification model was generated [24]. For the class predicted by the model for each image, the gradient of the last convolutional layer features for that class was identified to find the region that the classification model focuses on. Based on the regions of interest for each image, a more detailed picture of what information the classification model was classifying was generated for the image.

## 3. Results

### 3.1. The Performance of the Segmentation Network

The segmentation network has exhibited great performance compared with the ground truth labeled by retinal specialists, and the specific results of the segmentation task are demonstrated in Figure 2B and Appendix A. In layers 1–5, Dice scores were 0.997, 0.955, 0.896, 0.962, and 0.958, respectively. The ASD of the layers 1–5 were 0.014, 0.119, 0.308, 0.119, and 0.915, respectively. The mean Dice and ASD were 0.953 and 0.130, respectively, which demonstrated consistency with the ground truth in total and in the boundary.

### 3.2. The Comparison of Three Classification Networks

The performances of the three typical classification networks were compared in the internal dataset. The ResNet-34 architecture was noted to have the best results with a total accuracy of 81.9%, which was selected to be the backbone of the proposed two-stage network. In stage-specific performance, the areas under the curves (AUCs) of each category were 0.998, 0.924, 0.909, 0.904, 0.970, and 0.983. The detailed results are shown in Appendix A.

### 3.3. The Performance of iERM

The proposed two-stage architecture had a better performance compared with the traditional single-stage classification architecture. For the internal dataset, the AUCs of each category were increased to 0.999, 0.904, 0.925, 0.928, 0.974, and 0.990 with a total accuracy of 82.9% (Appendix A). For two external datasets, the total accuracy scores were improved from 84.0% to 87.0% in the Xian 1 dataset and from 73.5% to 79.4% to 79.4% in the Japanese dataset, respectively, with the assistance of segmentation (Table 3 and Table 4). The final AUCs of each category were 0.998, 0.826, 0.906, 0.958, 0.943, and 0.985 for the Xian 1 dataset and 0.999, 0.826, 0.906, 0.958, 0.943, and 0.985 for the Japanese dataset. The performance of fine-grained classification can be advanced by segmentation according to clinical prior knowledge. The receiver operating characteristic (ROC) curves and confusion matrices (CM) of the proposed two-stage system from two external datasets are illustrated in Figure 3 and Figure 4, respectively.

### 3.4. The Comparison of iERM and Human

Furthermore, four retinal ophthalmologists and four medical students were recruited to identify the stage of ERM for two external test datasets. The average total accuracy scores of four retinal ophthalmologists and four medical students were calculated and compared to the two-stage system. For the Xian 1 dataset, the average total accuracy scores of four ophthalmologists and four medical students were 87.8% and 79.4%, respectively (Figure 1E). For the Japanese dataset, those data were 81.2% and 54.1%. More detailed results are demonstrated in Appendix A. In conclusion, the DL system’s ability to diagnose and classify ERM was comparable to that of retinal ophthalmologists and performed better than medical students.

## 4. Discussion

This study has shown potential for AI-assisted fine-grained clinical grading. This study established an effective fine-grained grading system, that is, iERM, to directly guide ERM management. The proposed two-stage architecture, which implemented clinical prior knowledge segmentation as an auxiliary module, proved that it could improve the performance of the traditional classification model. This fine-grading system was evaluated with OCT image data collected from four different commercial OCT devices in multiple international medical centers, and its robustness and applicability were tested in two dependent external datasets. Furthermore, an ERM grading AI interactive interface was developed to show the progression of ERM and corresponding management recommendations applicable in clinical situations for patients.

This study substantially extends existing research and other studies in four aspects. First, ERM was classified into five severity grading subgroups by iERM. This subgroup categorization of ERM could clinically guide ERM management, including screening, referral, re-examination frequency, and surgical decision. According to previous research, decreasing best-corrected visual acuity of ERM patients had an association with the progression to more serious stages graded in OCT images, particularly progression from stage 2 to stage 3 [19]. The fine-grade ERM scheme could also provide a more objective and elaborate reference for clinical practices, rather than just detecting the presence of ERM. For example, patients with stage 5 ERM identified by iERM would be recommended for a referral to specialized retinal clinics regarding vitreoretinal surgery as soon as possible. Furthermore, this study established an iERM system with a user convenient interface (Appendix A). Patients or doctors can upload OCT reports and receive the analyzed results, including automatic diagnosis and management recommendations from iERM. Therefore, this AI system could contribute to the promotion of ophthalmological telemedicine. Additionally, it could improve access to medical services and reduce medical costs, particularly for distant regions. It could also alleviate ophthalmologists’ workload and reduce false-negative diagnosis.

Second, this proposed method integrated the segmentation network, with the classification network substantially increasing the results of automatic fine-grained clinical grading of ERM. Fine-grained clinical grading is a challenge for DL classification due to small inter-class variation and large intra-class variations [25]. To detect normal OCT images in the internal dataset, the Xian 1 dataset, and the Japanese dataset, the precisions were 98%, 100%, and 89.6%, with AUCs of 0.999, 0.998, and 0.999, respectively. This result showed the remarkable ability of iERM to distinguish normal OCT images from ERM images, which is effective and practical in terms of ERM screening. For fine-grained grading, ERM severity was based on OCT images. The similarity of specific grades was quite high compared to junior ophthalmologists or ophthalmologists, who might not be able to tell the difference due to lack of experience. The single-stage classification network achieved total accuracy scores of 81.9%, 84%, and 73.5% for the internal dataset, the Xian 1 dataset, the Japanese dataset, respectively. In Figure 1D, with the assistance of a segmentation network during two-stage architecture, total accuracy scores were noted to improve to 82.9%, 87.0%, and 79.4%, respectively.

The final novelty of this study was the implementation of segmentation with prior clinical knowledge that demonstrated the interpretability of this DL model. The transparency of the DL is an underlying issue, which could impede the application of AI in clinical practice (22, 23). Previous DL studies have integrated the segmentation network with the classification network to tackle the “black box” problem [26,27,28,29]. Jeffrey et al. first trained a DL network to create a segmentation map of 15 lesions or elements and then utilized it as input for second stage classification [26]. The segmentation map provided an interpretable representation for doctors, which was particularly practical in difficult and ambiguous cases. Xu et al. established a hierarchical DL system with prior human knowledge to solve this problem [27]. Extraction and use of anatomical characteristics of the fundus images comprehensively simulated the diagnostic thinking of human experts. In this study, the specific layer of segmentation was designed according to the clinical ERM severity grade standard. The key features of grading ERM progression were the morphological changes of retinal tissues displayed in OCT images. From a retinal specialists’ perspective, clinical diagnosis of ERM based on OCT images was also a two-stage process. First, it requires discovery and analysis of the morphological changes of specific key features and then synthesis of all information to make a diagnosis decision. This study’s proposed method adopted this two-stage clinical diagnosis strategy and was proven to be effective. It is more interpretable for ophthalmologists than the conventional DL classification systems.

The last main contribution of this work is the application of diverse datasets obtained from multiple OCT devices of multiple international medical institutes. The generalizability of DL algorithms has been a concern in multi-center studies [30]. In Figure 1B, the presentation of OCT images was diverse from various OCT devices or various capture patterns of the same OCT device. Though those OCT images displayed similar lesions of the same disease, the different grayscale information resulted in the heterogeneity of the data, which could also create an impediment for the applicability of DL algorithms in a real-world setting. In this study, the DL algorithm was correlated with OCT data from four common commercial devices and tested it on two independent external datasets. This work attempts to bridge the gap of data heterogeneity.

This current study had several limitations. First, the dataset size of OCT images from the other OCT device was relatively small, compared to the Heidelberg device. Thus, large-scale studies should be performed, and an external dataset collected from diverse OCT devices should be created to further test the generalizability of this DL system in the future. Second, the image number of each sub-category was imbalanced due to various prevalence rates of specific stages. Even though this study attempted to adopt some preprocessing methods, the data imbalance could also influence the performance. Finally, the interpretability seemed still insufficient to be applied in clinical practice. It may hinder trust from healthcare providers and result in an important challenge for the integration of AI systems in clinical settings. Joint efforts should be made by healthcare providers and AI developers in future work.

## 5. Conclusions

This study proved a benchmark method to improve the performance and enhance the interpretability of the traditional DL model with the implementation of segmentation based on prior human knowledge. This study successfully proposed a two-stage DL framework for the fine-grading assessment of ERM progression and achieved advanced results comparable to retinal specialists based on multi-source data, which contributed to efficient clinical ERM management and automatic diagnosis of retinal diseases.

## Figures and Tables

**Figure 1 jcm-12-00400-f001:**
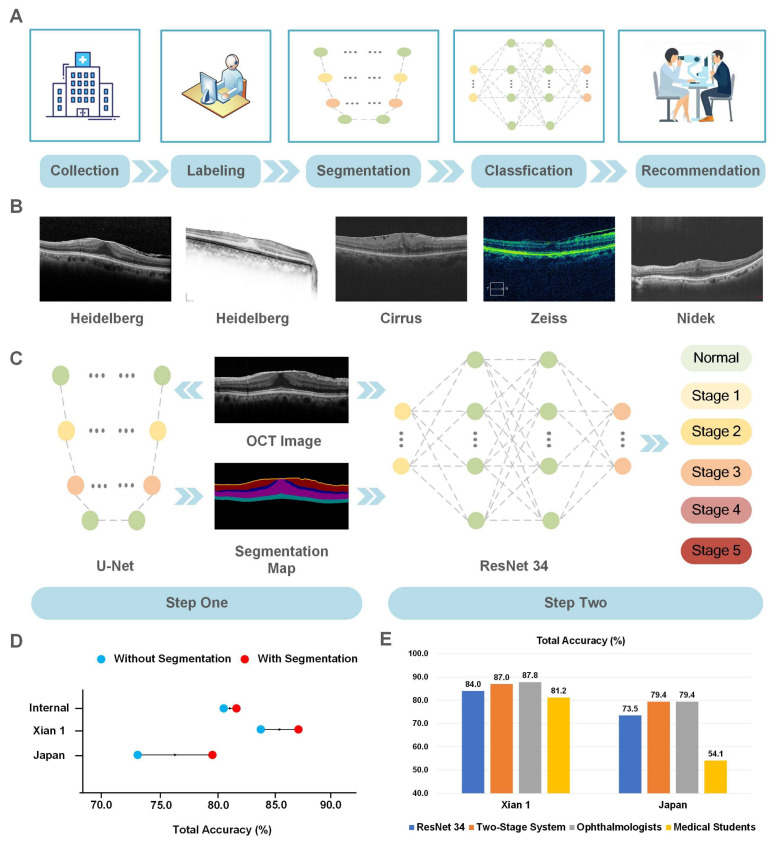
Overview of iERM for ERM severity grading through OCT images. (**A**) Workflow of the iERM system. (**B**) The examples of OCT data collected from different commercial devices. (**C**) A brief demonstration of the proposed two-stage system and the schematic diagram of deep learning networks. (**D**) The performance of ERM severity grading on the internal dataset and two external datasets was improved by the two-stage system, compared to the traditional classification model. (**E**) Comparison of the average total accuracy scores of four retinal ophthalmologists and four medical students in two external datasets with those corresponding to the two-stage system and ResNet-34.

**Figure 2 jcm-12-00400-f002:**
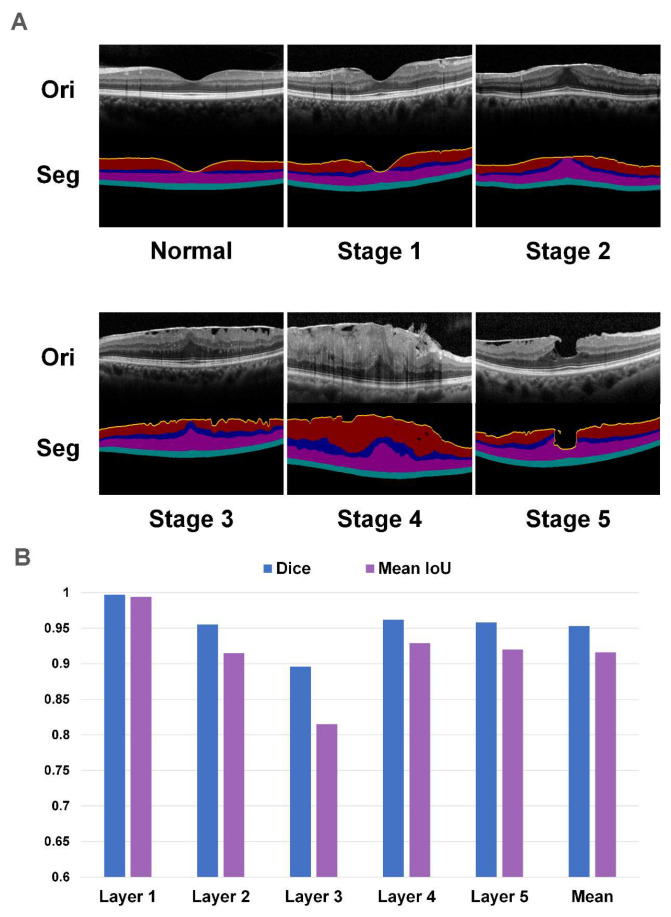
The demonstration of classification standard and segmentation results. (**A**) The typical case of each stage and their corresponding segmentation maps. (**B**) The Dice coefficient (Dice) scores and mean intersection over union (IoU) of layers 1–5. Ori: the original OCT images. Seg: the segmentation maps.

**Figure 3 jcm-12-00400-f003:**
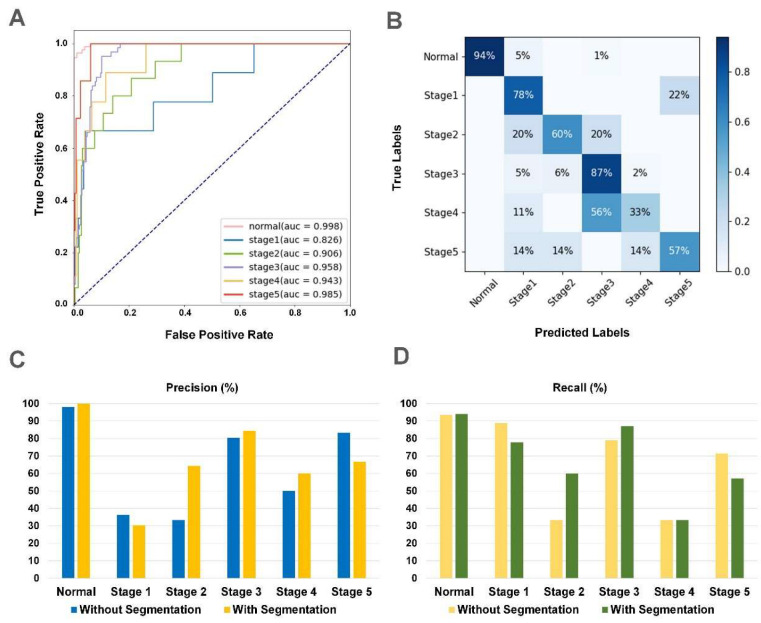
The results of iERM in the Xian 1 dataset. (**A**) The receiver operating characteristic (ROC) curve of iERM. (**B**) The confusion matrices (CM) of iERM. (**C**) Comparison of iERM with the traditional classification model on the precision of each stage. (**D**) Comparison of iERM with the traditional classification model for the recall of each stage.

**Figure 4 jcm-12-00400-f004:**
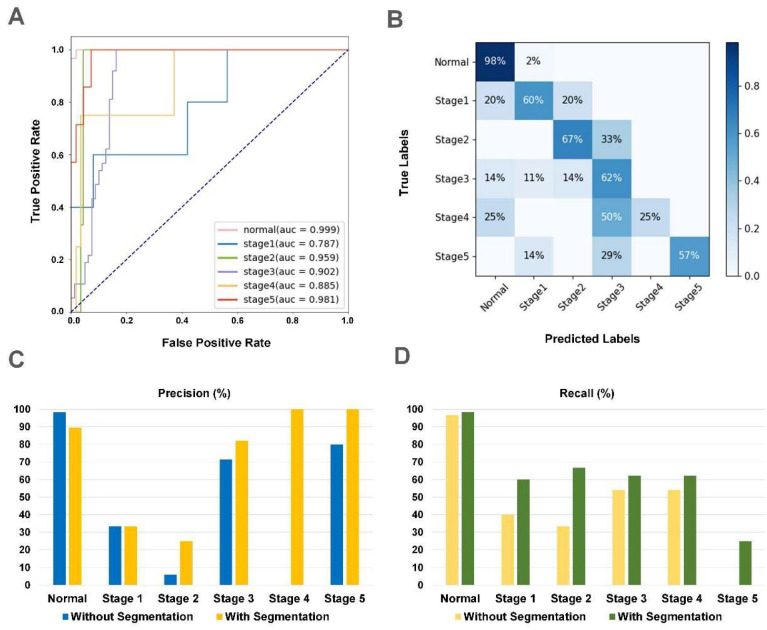
The results of iERM in the Japanese dataset. (**A**) The receiver operating characteristic (ROC) curve of iERM. (**B**) The confusion matrices (CM) of iERM. (**C**) Comparison of iERM with the traditional classification model on the precision of each stage. (**D**) Comparison of iERM with the traditional classification model on the recall of each stage.

**Table 1 jcm-12-00400-t001:** The demographics and statistics information of patients in the ZJU dataset.

	Age (Year)	Male Sex (%)	OD/OS	OCT Images
Normal	61.52 ± 7.90	44.6	0.96	1674
Stage 1	63.12 ± 9.13	49.4	0.98	288
Stage 2	64.35 ± 7.98	44.1	1.02	440
Stage 3	65.52 ± 8.71	47.6	0.96	741
Stage 4	65.14 ± 6.34	45.9	1.03	185
Stage 5	65.47 ± 7.83	52.0	1.03	229
Total	63.29 ± 8.29	46.0	0.98	3557

**Table 2 jcm-12-00400-t002:** The data distribution of all datasets.

Dataset	OCT Device	Image Number	Normal	Stage 1	Stage 2	Stage 3	Stage 4	Stage 5
ZJU	Heidelberg	3557	1674	288	440	741	185	229
Xian 1	Heidelberg	270	168	9	15	62	9	7
Ningbo	Heidelberg	84	17	10	16	35	3	3
Jinhua	Heidelberg	126	40	12	11	42	11	10
Dali	Heidelberg	121	0	2	6	69	38	6
Anhui	Heidelberg	19	0	1	0	13	3	2
Japan	Heidelberg	117	61	5	3	37	4	7
Singapore 1	Heidelberg	71	8	5	17	19	5	17
Taizhou	Nidek	22	0	5	4	12	1	0
Xian 2	Zeiss	109	0	16	10	49	26	8
Singapore 2	Cirrus	78	0	21	12	23	8	14

**Table 3 jcm-12-00400-t003:** The results of iERM in the Xian 1 dataset compared to the traditional classification model.

Xian 1	Normal	Stage 1	Stage 2	Stage 3	Stage 4	Stage 5
without segmentation						
Precision (%)	98.1	36.4	33.3	80.3	50.0	83.3
Recall (%)	93.5	88.9	33.3	79.0	33.3	71.4
F1-score (%)	95.7	51.6	33.3	79.7	40.0	76.9
AUC (%)	99.4	86.3	81.7	97.1	89.7	97.1
Accuracy (%)			84.0			
with segmentation						
Precision (%)	100.0	30.4	64.3	84.4	60.0	66.7
Recall (%)	94.0	77.8	60.0	87.1	33.3	57.1
F1-score (%)	96.9	43.7	62.1	85.7	42.9	61.5
AUC (%)	99.8	82.6	90.6	95.8	94.3	98.5
Accuracy (%)			87.0			

**Table 4 jcm-12-00400-t004:** The results of iERM in the Japanese dataset compared to the traditional classification model.

Japan	Normal	Stage 1	Stage 2	Stage 3	Stage 4	Stage 5
without segmentation						
Precision (%)	98.3	33.3	5.9	71.4	0.0	80.0
Recall (%)	96.7	40.0	33.3	54.1	0.0	57.1
F1-score (%)	97.5	36.4	10.0	61.5	0.0	66.7
AUC (%)	96.5	72.3	83.3	90.9	84.5	96.0
Accuracy (%)			73.5			
with segmentation						
Precision (%)	89.6	33.3	25.0	82.1	100.0	100.0
Recall (%)	98.4	60.0	66.7	62.2	25.0	57.1
F1-score (%)	93.7	42.9	36.6	70.8	40.0	72.7
AUC (%)	99.9	78.7	95.9	90.2	88.5	98.1
Accuracy (%)			79.4			

## Data Availability

The de-identified individual participant data can be requested from the corresponding author, who will evaluate such requests on a case-by-case basis.

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
