# Peer review of "iERM: An Interpretable Deep Learning System to Classify Epiretinal Membrane for Different Optical Coherence Tomography Devices: A Multi-Center Analysis"

_jcm, 2023, doi:10.3390/jcm12020400_

Round 1

Reviewer 1 Report

The authors have successfully used DL with image labeling for more accurate ERM staging. In order to increase the accuracy of DL, we believe that image weighting such as this one is important and essential for future clinical applications.

Please check the following points as MINOR corrections.
・In Fig. 1 and Table 2, NIDEK is listed as NEDIK, which should be corrected.
・In the discussion, it was stated that stage 5 is desirable to have surgery immediately, but this needs to be corrected as it is considered to be clinically different

Author Response

Response to Reviewer 1 Comments

Reviewer 1 comments: The authors have successfully used DL with image labeling for more accurate ERM staging. In order to increase the accuracy of DL, we believe that image weighting such as this one is important and essential for future clinical applications.

Please check the following points as MINOR corrections.

・In Fig. 1 and Table 2, NIDEK is listed as NEDIK, which should be corrected.

・In the discussion, it was stated that stage 5 is desirable to have surgery immediately, but this needs to be corrected as it is considered to be clinically different

Overall response: We quite appreciate your detailed reviewing and meaningful suggestions. We have studied your comments and revised the paper according to your comments in a point-to-point way. The point-to-point responses are provided below. The key points of our responses are marked in bold.

Point 1: In Fig. 1 and Table 2, NIDEK is listed as NEDIK, which should be corrected.

Response 1: Thanks for your kind suggestion considering the mistakes. We have rechecked the manuscript and revised these errors.

Point 2: In the discussion, it was stated that stage 5 is desirable to have surgery immediately, but this needs to be corrected as it is considered to be clinically different

Response 2: Thanks for your suggestion. We agree with you that the conclusion should be demonstrated more accurately. We have altered our expression in Paragraph 2. Discussion (Line 37-38):

“For example, patients with stage 5 ERM identified by iERM would be recommended for a referral to specialized retinal clinics regarding vitreoretinal surgery as soon as possible.”

Reviewer 2 Report

The authors of this study have studied an important disease entity i.e. ERM. They have developed an ERM grading AI interactive interface called iERM, which they recommend adapting for the efficient management of ERM. The methodology and results are well described.

Author Response

Response to Reviewer 2 Comments

Reviewer 2 comments: The authors of this study have studied an important disease entity i.e. ERM. They have developed an ERM grading AI interactive interface called iERM, which they recommend adapting for the efficient management of ERM. The methodology and results are well described.

Response: We quite appreciate your detailed reviewing and meaningful suggestions. Those comments are very valuable and helpful for the improvement of our paper. We have studied the comments carefully and have revised the paper according to the reviewers’ comments. And the revised portion has been by yellow highlighting.

Reviewer 3 Report

This study provides the technical and clinical considerations in building DL algorithms for ERM diagnosis. Especially, it is a noteworthy achievement that the interpretability of the traditional model was improved using prior clinical knowledge. 

However, the interpretability is still insufficient to be applied in clinical practice. Please briefly discuss future direction to improve its peformance. 

Author Response

Response to Reviewer 3 Comments

Reviewer 3 comments: This study provides the technical and clinical considerations in building DL algorithms for ERM diagnosis. Especially, it is a noteworthy achievement that the interpretability of the traditional model was improved using prior clinical knowledge.

However, the interpretability is still insufficient to be applied in clinical practice. Please briefly discuss future direction to improve its peformance.

Overall response: We quite appreciate your detailed reviewing and meaningful suggestions. We have studied your comments and revised the paper according to your comments in a point-to-point way. The point-to-point responses are provided below. The key points of our responses are marked in bold.

Point 1: The interpretability is still insufficient to be applied in clinical practice. Please briefly discuss future direction to improve its peformance.

Response 1: Thanks for your insightful opinion and kind suggestion. We would love to briefly discuss future direction to improve its peformance in the last paragraph of Discussion (Line 84-86).

“Last, the interpretability seemed still insufficient to be applied in clinical practice. It may hinder trust from healthcare providers and result in an important challenge for the integration of AI systems in clinical settings. Joint efforts should be made by healthcare providers and AI developers in future work.”

Reviewer 4 Report

In this paper, the authors aimed to develop a two-stage deep learning (DL) system named iERM to provide accurate automatic grading of ERM for clinical practice. Overall, I found the paper well written, clinical data is clearly presented and statistical analyses seem sound. There are many values in the work presented, the main drawback of this study is that the dataset size of OCT images from the other OCT device was relatively small, compared to the Heidelberg device which authors have mentioned as a limitation. 

My decision: Minor revision.

1-In the material and methods section: “The workflow of the iERM system was demonstrated in Figure” Please write more details about the steps

2-It was better to test the accuracy, robustness, and applicability on a separate data set of Zeiss images and Heidelberg images or discuss this issue.

Author Response

Response to Reviewer 4 Comments

Reviewer 4 comments: In this paper, the authors aimed to develop a two-stage deep learning (DL) system named iERM to provide accurate automatic grading of ERM for clinical practice. Overall, I found the paper well written, clinical data is clearly presented and statistical analyses seem sound. There are many values in the work presented, the main drawback of this study is that the dataset size of OCT images from the other OCT device was relatively small, compared to the Heidelberg device which authors have mentioned as a limitation.

My decision: Minor revision.

1-In the material and methods section: “The workflow of the iERM system was demonstrated in Figure” Please write more details about the steps

2-It was better to test the accuracy, robustness, and applicability on a separate data set of Zeiss images and Heidelberg images or discuss this issue.

Overall response: We quite appreciate your detailed reviewing and meaningful suggestions. We have studied your comments and revised the paper according to your comments in a point-to-point way. The point-to-point responses are provided below. The key points of our responses are marked in bold.

Point 1: 1-In the material and methods section: “The workflow of the iERM system was demonstrated in Figure” Please write more details about the steps.

Response 1: Thanks for your suggestion. We have supplemented this information in the material and methods section.

“The workflow of the iERM system was demonstrated in Figure 1(a), which included data collection, image labelling, development of the segmentation model, and development of the classification model.”

Point 2: It was better to test the accuracy, robustness, and applicability on a separate data set of Zeiss images and Heidelberg images or discuss this issue.

Response 2: Thanks for your suggestion. We tried to test the accuracy, robustness, and applicability on a separate data set of Zeiss images and Heidelberg images, but finally resulted in confusion because the image size of Zeiss is too small comparing to Heidelberg. We have discussed this point in the last paragraph of Discussion (Line 79-81).

“First, the dataset size of OCT images from the other OCT device was relatively small, compared to the Heidelberg device. Thus, large-scale studies should be performed, and an external dataset collected from diverse OCT devices should be created to further test the generalizability of this DL system in the future.”
